# Common Oral Medications Lead to Prophage Induction in Bacterial Isolates from the Human Gut

**DOI:** 10.3390/v13030455

**Published:** 2021-03-11

**Authors:** Steven G. Sutcliffe, Michael Shamash, Alexander P. Hynes, Corinne F. Maurice

**Affiliations:** 1Department of Microbiology and Immunology, McGill University, Montreal, QC H3A2B4, Canada; steven.sutcliffe@mail.mcgill.ca (S.G.S.); michael.shamash@mail.mcgill.ca (M.S.); 2Department of Medicine, McMaster University, Hamilton, ON L8S4L8, Canada; hynes@mcmaster.ca

**Keywords:** bacteriophage, gut microbiota, lysogeny, prophage, induction, medication, antimicrobial

## Abstract

Many bacteria carry bacteriophages (bacterial viruses) integrated in their genomes in the form of prophages, which replicate passively alongside their bacterial host. Environmental conditions can lead to prophage induction; the switching from prophage replication to lytic replication, that results in new bacteriophage progeny and the lysis of the bacterial host. Despite their abundance in the gut, little is known about what could be inducing these prophages. We show that several medications, at concentrations predicted in the gut, lead to prophage induction of bacterial isolates from the human gut. We tested five medication classes (non-steroidal anti-inflammatory, chemotherapy, mild analgesic, cardiac, and antibiotic) for antimicrobial activity against eight prophage-carrying human gut bacterial representative isolates in vitro. Seven out of eight bacteria showed signs of growth inhibition in response to at least one medication. All medications led to growth inhibition of at least one bacterial isolate. Prophage induction was confirmed in half of the treatments showing antimicrobial activity. Unlike antibiotics, host-targeted medications led to a species-specific induction of *Clostridium beijerinckii*, *Bacteroides caccae*, and to a lesser extent *Bacteroides eggerthii*. These results show how common medication consumption can lead to phage-mediated effects, which in turn would alter the human gut microbiome through increased prophage induction.

## 1. Introduction

The human gut is at the intersection of host cells, trillions of microorganisms (bacteria, archaea, eukaryotes, and viruses), and all the different compounds we ingest, termed xenobiotics. The bacterial fraction of this microbial community is responsible for the metabolism of a wide range of xenobiotics, including components of our diet [1]. The increase in medication consumption in the United-States [2] and globally [3] makes medication an important xenobiotic shaping our gut microbiota. Medication of a variety of classes can have major effects on the gut bacteriome [4,5,6,7,8,9,10,11,12] leading to species-specific bacterial growth inhibition [13] or community-level shifts in bacterial diversity [4,5,6,7,8,9,10,11,12].

Medication can also alter the gut virome [14], which is highly correlated with the bacterial community [15]. This is because the gut virome is dominated by bacteriophages [16] (phages): viruses that infect and lyse bacteria. The majority of phages in the gut are identified as temperate [17,18,19], meaning they are capable of replicating lysogenically. Lysogenic replication includes the incorporation of the phage genome into the host bacterial genome as a prophage (or as a plasmid) [20]. Bacterial hosts with prophages are termed lysogens. Prophages are found in about half of bacterial isolates [21] and commonly found in complex communities [22], including the murine [23] and human gut [24]. Prophages are not simply hitchhiking genetic cargo, but play an important ecological role in the gut through super-infection immunity [25], lysogenic conversion or transduction [26], and encode genes involved in a number of processes associated with anaerobic respiration, as well as genes involved in amino acid, carbohydrate, nucleotide, lipid, and even xenobiotic metabolism [18].

Lysogeny is not a static state: prophages contain molecular switches that allow for the return to lytic replication, a process referred to as prophage induction. Prophage induction is likely an important driver of phage-bacteria dynamics in the gut. For example, in Crohn’s disease patients, the shifts in gut virome diversity appear to be caused by prophage induction [27]. Determining the role of prophage induction as a driver of phage–bacteria dynamics in the gut requires identifying the conditions that trigger prophage induction first. 

Prophage induction is typically triggered through bacterial DNA-damage. Work with bacterial isolates and clinical observations suggest RecA activation by antibiotics leads to prophage induction in situ [28]. Other xenobiotics such as specific dietary compounds [29] dietary fructose, and short-chain-fatty acids [30] have been shown to induce gut lysogens, and whole diet changes have also been shown to alter both murine [23] and human gut virome diversity [14,19]. Non-antibiotic medications are also likely inducers, as they are capable of inhibiting gut bacterial growth through a variety of mechanisms [13], correlating with gut virome variation [14] and an up-regulation of phage genes in the gut bacterial community [12]. We thus hypothesize that many oral medications, including non-antibiotics, induce prophages of human gut lysogens.

We screened a variety of medications for prophage induction against lysogenic human gut bacterial isolates. We quantified virus-like-particles (VLP) by epifluorescence microscopy and confirmed prophage induction of in silico predicted prophages. Our results confirm that bacterial growth inhibition by medications, including non-antibiotic drugs, leads to an increase in phages through prophage induction, and could be altering the virome and resulting in phage-mediated shifts in the gut microbiome.

## 2. Materials and Methods

### 2.1. Bacterial Isolates

We selected eight human gut bacterial isolates: Bacteroidetes (*Bacteroides caccae*, *Bacteroides ovatus*, *Bacteroides eggerthii*), Firmicutes (*Clostridium beijerinckii*, *Clostridium scindens*, *Enterococcus faecalis*), Proteobacteria (*Escherichia coli*), and Actinobacteria (*Bifidobacterium longum* subsp. *infantis*). All isolates are associated with the human gut microbiota (Table 1) and represent the major phyla of the gut [31]. All of our tested isolates had genomes assembled, at least at a scaffold level, with exception of *E. coli* and *C. scindens* (Table 1). *E. coli* and *C. scindens* were shown to be lysogens experimentally, and the rest were determined to be lysogens based on prophage prediction. Prophages were predicted on the bacterial genomes using PHASTER [32] and VirSorter (Appendix A) [33].

### 2.2. Estimation of Medication Concentrations in the Human Gut

Information on the concentration of medications selected in the human gut is currently unavailable. We first selected medications that are taken orally, as they are likely to interact with the human gut microbiota [34]. The human gut contains bacteria along the entire gastrointestinal tract but is in highest density and diversity in the large intestine [35]. The colon is the site of most gut microbiota studies, specifically the lumen [36]. Orally administered medications rarely target the colon as the site of action and most of the absorption occurs earlier in the small intestine. The amount absorbed and found in the circulatory system, or bioavailability, is therefore well studied. We estimated the concentration in the colon of our tested medications based on loss of oral dose by bioavailability (Appendix A). This model does not take into account medications entering the gut through biliary excretion or in a transformed state but is an estimate for the concentration found in the gut.

### 2.3. Preparation of Medication

Stock solutions were made with powdered medications (ampicillin sodium salt, (A0166) CAS: 69-52-3; ciprofloxacin, (17850) CAS: 85721-33-1; norfloxacin, (N9890) CAS: 70458-96-7; diclofenac sodium salt, (D6899) CAS: 15307-79-6; ibuprofen, (14883) CAS: 15687-27-1; tolmetin, (1670502) CAS: 64490-92-2; digoxin, (D6003-1G) CAS: 20830-75-5; streptonigrin from *Streptomyces flocculus* (S1014) CAS: 3930-19-6; busulfan, (B2635) CAS: 55 98-1; fludarabine phosphate, USP (1272204) CAS: 75607-67-9 Sigma-Aldrich Canada Co., Oakville, ON, Canada; mitomycin C, (BP253110) CAS: 50-07-7 Fisher Scientific, Nepean, ON, Canada) dissolved in dimethyl sulfoxide (DMSO) to a concentration of 10 mg mL^−1^, except where solubility did not permit, for ciprofloxacin (0.2 mg mL^−1^), streptonigrin, norfloxacin, and tolmetin (1 mg mL^−1^), and stored at −20 °C. DMSO was chosen as a solvent due to its ability to dissolve non-antibiotics (fludarabine, ibuprofen, and diclofenac) that have low solubility in water. Medications were serially diluted in DMSO such that 2 μL added to 200 μL wells had final concentrations of 0.01, 0.10, 1.00, 10.00, and 100 μg mL^−1^ (with the exception of previous low solubility medications) in media. This was to reduce DMSO concentration in media, as it can inhibit bacterial growth at high concentrations. In addition, we tested a higher concentration of ciprofloxacin dissolved in slightly acidic water (pH 6.5, final concentration 2 mg mL^−1^) on a subset of bacterial isolates that did not show induction at the lower tested concentrations.

### 2.4. In Vitro Treatments

We grew all bacteria anaerobically (Coy chamber with 5% hydrogen, 20% carbon dioxide, 95% nitrogen, Mandel Scientific Company Inc., Guelph, ON, Canada). Simulating the human gut environment temperature at 37 °C in nutrient rich environment with general fastidious growth broth (brain heart infusion broth (BHI) BBL 299070 BD, Mississauga, ON, Canada, anaerobe basal broth (ABB) CM0957, Thermo Scientific, Waltham, MA, USA, tryptic soy broth No. 2 (TSB) 51288 Millipore, Oakville, ON, Canada with or without 0.1% hemin chloride in NaOH (5 mg mL^−1^) (Table 1). Bacteria were grown in 96-well plates, measuring OD_600 nm_ by spectrophotometry (Epoch 2 microplate spectrophotometer, Biotek Instruments, Winooski, VT, USA) and mixing every five min until early exponential phase (1/4 OD of stationary phase). At the early exponential phase, medications dissolved in DMSO were added (2 μL) to reach their tested concentration (*n* = 3) along with DMSO control (*n* = 3). Bacterial growth was then monitored with an OD_600 nm_ reading/mixing every 15 min until stationary phase (~24 h). Slow growing bacteria (*B. caccae*, *B. ovatus*) were grown for ~48 h and faster growing bacteria (*E. coli*) ~8 h. Then, 96-well plates were fixed with w.v 2% formaldehyde and stored at −20 °C for VLP enumeration. The area under the growth curve (AUC) was calculated after medications were administrated and calculated with Prism (version 7, GraphPad Software, San Diego, CA, USA). AUC for each treatment was calculated compared to the DMSO control for each bacterium on the day of their induction (*n* = 3). AUC decreases >15% were investigated for VLP production. 

### 2.5. VLP Enumeration

Fixed samples were from the 96-well plates were thawed and centrifuged at 2000× *g* for 20 min. The VLP-containing supernatant was collected on 0.02 µm Whatman Anodisc filters (GE Healthcare, Chicago, IL, USA) and stained with 2.5 × SYBR Gold stain (final concentration, ThermoFisher Scientific, Waltham, MA, USA) before enumeration on an Axioskop (Zeiss, Oberkochen, Germany) epifluorescence microscope at 1000X. We counted a minimum of 300 events per slide, or 30 regions to increase statistical power of counts.

### 2.6. Prophage Induction of C. beijerinckii for DNA Sequencing and PCR

We performed increased in silico prophage prediction on the strain of *C. beijerinckii* with additional computational tools (VIBRANT [37] and PhiSpy [38]), and it was shown to contain eleven unique putative prophage regions. ORFs of the putative prophage regions were predicted and annotated with HMMER (v3.2.1) [39] and the pVOG (version May 2016) database [40]. Annotated ORFs of putative prophage regions were grouped based on belonging to five functional modules (lysogeny, genome replication, head morphogenesis, tail morphogenesis, and host lysis). 

PCR primers (Appendix A) were designed for all the complete and uncertain regions (P1, P2, P3, P4, P5, P7, P10) in addition to a bacteria-specific primer for the *C. beijerinckii dnaA* gene. We tested all primers on bacterial gDNA and confirmed their specificity with Sanger sequencing. To generate larger quantities of unfixed VLPs we repeated our induction protocol for ciprofloxacin 2 µg mL^−1^, mitomycin 1 µg mL^−1^, norfloxacin 10 µg mL^−1^, and ampicillin 0.1 µg mL^−1^ in 42 wells of a PCR plate to increase the volume of sample.

### 2.7. Purification of Viral DNA from VLPs

Phage supernatants were concentrated by centrifugation. The phage pellet was resuspended in SM buffer (100 mM NaCl, 8 mM MgSO_4_·7 H_2_O, 50 mM Tris-Cl (pH 7.5)) and incubated sequentially with lysozyme (50 mg mL^−1^), TURBO DNase and TURBO DNase buffer (ThermoFisher Scientific, Waltham, MA, USA), and proteinase K (20 mg mL^−1^). Then, 5 M NaCl and 10% CTAB/0.7 M NaCl solution were added, and samples were transferred to phase lock gel tubes (light PLG tubes, QuantaBio, Beverly, MA, USA) with an equal amount of phenol:chloroform:isoamyl alcohol (25:24:1 *v/v*, pH = 8.0, ThermoFisher Scientific, Waltham, MA, USA) and centrifuged. The top aqueous DNA-containing layer was left to precipitate overnight at −80 °C in 100% ice-cold ethanol and samples were then purified with the Zymo DNA Clean and Concentrator 25 kit (Zymo Research, Irvine, CA, USA). DNA concentrations were quantified with the Qubit dsDNA high-sensitivity (HS) assay kit (ThermoFisher Scientific, Waltham, MA, USA).

### 2.8. Extraction of Genomic DNA from Gut Bacterial Isolates

Bacterial genomic DNA was then extracted using the Qiagen DNeasy Blood and Tissue kit (Qiagen, Germany) and concentrated with the Zymo DNA Clean and Concentrator 100 kit (Zymo Research, Irvine, CA, USA), as per the manufacturers’ instructions. DNA concentrations were quantified with the Qubit dsDNA broad-range (BR) assay kit (ThermoFisher Scientific, Waltham, MA, USA).

### 2.9. Shotgun Sequencing of Purified Viral DNA & Processing of Sequencing Data

Purified vDNA from each experiment was sheared using a Covaris ultrasonicator (Covaris, Woburn, MA, USA) and dual-indexed paired-end Illumina sequencing libraries were prepared using the Accel-NGS 1S Plus kit (Swift Biosciences, Ann Arbor, MI, USA). Pooled libraries were sequenced with 250 bp paired-end sequencing technology on the Illumina HiSeq platform at the Swift Biosciences facility and then trimmed with Trimmomatic (v0.83) [41]. Trimmed quality-filtered reads were aligned to the corresponding reference bacterial chromosome with Bowtie2 (v2.3.4.3) [42]. Manual curation of read coverage along the bacterial chromosome was done in Geneious Prime (v2020.0.4; Biomatters). The mean coverage of a given prophage region was calculated using the “bedcov” command in SAMtools. Mean coverage was normalized to the number of filtered reads in the sample, an approach known as total-sum scaling [43]. The “coverage” command in bedtools (v2.29.0) was used to determine the number of reads mapping to each prophage region within a given sample [44]. Circleator (v1.0.2) was used to generate figures containing bacterial genomes annotated with %GC content and the annotated predicted prophage regions [45].

## 3. Results

### 3.1. In Vitro Model to Study Prophage Induction of Human Gut Bacteria

We screened 480 different conditions for bacterial inhibition: 12 medications at five different concentrations for each of our eight bacterial isolates. We selected four categories of medications reported to impact human gut bacteria: non-steroidal anti-inflammatory (NSAID; diclofenac, ibuprofen, tolmetin) [9,10], chemotherapy (busulfan, fludarabine) [11], cardiac medications (digoxin) [12], and antibiotics (ampicillin, ciprofloxacin, norfloxacin, streptonigrin, mitomycin) [4,5,6,7,8], along with acetaminophen, the most commonly used analgesic (Table 2) [46]. All of the medications chosen are taken orally, which is more relevant to the human gut microbiota than intravenous medications [34]. Diclofenac and ibuprofen have been previously reported to inhibit growth of bacterial isolates [47,48,49]. Fludarabine and digoxin have been shown to inhibit growth of human gut bacteria in conditions relevant to the human gut [13]. Fludarabine was also shown to exhibit increased cytotoxicity in the presence of bacteria [50]. Ciprofloxacin, ampicillin, digoxin, and norfloxacin led to differential expression of gut bacterial genes, some of which were related to phage replication [12]. Antibiotics were selected based on their reported ability to induce prophages [51,52,53,54].

A wide range of concentrations relevant to the gut microbiota were tested as prophage induction can occur between maximum and minimum bacterial inhibition concentrations [55]. In the absence of data on the concentrations of our tested medications in the gut or in faeces, we estimated colon concentrations using the common oral dosage and the bioavailability of each medication, with the exception of mitomycin (Appendix A). Tested medication concentrations (Table 2) were determined to be physiologically relevant to the human gut microbiota: half the medications had at least one tested concentration that fell within the range of estimated colon concentrations, the other half tested were below the estimated colon concentration (Table 2). The median estimated concentration in the colon of our tested medications was 86.90 μg mL^−1^, below our maximum tested concentration of 100 μg mL^−1^. We limited our study to the relevant medication concentrations in an effort to approximate in vitro conditions to that of the human gut.

### 3.2. Antibacterial Activity of Medications on Human Gut Isolates In Vitro

Inhibition of bacterial growth can either be caused by the direct antibacterial effect of the medication, or by cell lysis from prophage induction. Here, we used inhibition of bacterial growth as a preliminary screen of 480 different treatments which may lead to prophage induction.

Antibacterial activity was measured by the difference in the AUC between the control (DMSO) and the treatment (Figure 1A). Bacterial growth inhibition was defined here by an antibacterial activity that leads to a decrease in the AUC of 15% or more (AUC15). Of the 480 treatments tested, 64 (13%) led to bacterial growth inhibition (Figure 1B). All of our bacterial isolates were inhibited by at least one medication at one concentration tested, except *E. faecalis* (Figure 1B). As predicted, antibiotics led to the most treatments with bacterial growth inhibition, specifically ampicillin and mitomycin, inhibiting five and seven bacteria, respectively (Figure 1B).

All non-antibiotic medications were able to inhibit the growth at least one bacterial isolate. However, only three of the eight bacterial isolates (*B. eggerthii*, *B. caccae*, and *C. beijerinckii*) were inhibited by non-antibiotics, with *B. caccae* and *C. beijerinckii* making up 14 of 15 the cases of inhibition (Figure 1B). Diclofenac (100 μg mL^−1^) and tolmetin (10 μg mL^−1^) were the only non-host-targeted medications to lead to a decrease of greater than 50% (AUC50) (Figure 1B).

### 3.3. Medication Caused Prophage Induction of Human Gut Lysogens

We defined prophage induction as the combination of bacterial growth inhibition (Figure 1) and a significant increase in VLP compared to control (Figure 2A). We thus further studied the 64 treatments leading to the inhibition of bacterial growth, spanning all 12 tested medications, for changes in VLPs (Figure 2B), as counted by epifluorescence microscopy (Figure 2A).

Most bacterial isolates with growth inhibition had a corresponding increase in VLPs (84%), and over half (55%) increased significantly compared to the controls (Figure 2B), indicating prophage induction. Prophage induction is isolate- and medication-specific: no one medication induced all inducible prophages, and on average, bacteria were induced by three different medications (rarely the same ones), with results often concentration specific (Figure 2B).

Mitomycin, a commonly used prophage inducer for lysogeny estimates and prophage detection [56,57] was our most widespread inducer as expected, resulting in the lysis of five of eight strains and representing approximately one third of treatments where induction occurred (Figure 2B). Only *B. caccae* was not inhibited by mitomycin, despite containing an inducible prophage (Figure 2B). Ciprofloxacin is also a common antibiotic for prophage induction, yet it did not inhibit many bacteria at the low concentration we tested (Figure 1B). We thus increased its concentration to 20 μg mL^−1^ by dissolving in slightly acidic water (pH 6.5) and tested the non-induced inhibited bacteria with this higher concentration. All bacteria tested with the higher ciprofloxacin concentration were inhibited, but only *C. scindens* was lysed as a result of prophage induction (Appendix A).

Ten of our twelve tested medications led to prophage induction, including five host-targeted medications, spanning all the medication categories: diclofenac (NSAID), tolmetin (NSAID), fludarabine (chemotherapy), acetaminophen (analgesic), digoxin (cardiac) (Figure 2B). Diclofenac was the only non-antibiotic to cause induction in more than one bacterial isolate (*B. caccae* and *B. eggerthii*). *B. caccae* and *C. beijerinckii* make up more than half of the positive results for non-antibiotic prophage induction. This indicates that specific gut isolates are more susceptible to non-antibiotic medications. Only two non-antibiotics did not lead to prophage induction in our isolates: ibuprofen (NSAID) and busulfan (chemotherapy), despite increasing overall VLP counts (3-fold for ibuprofen, adjusted *p*-value: 0.384; 3-fold increase for busulfan, adjusted *p*-value: 0.983; Figure 2B).

### 3.4. Confirmation of In Silico Predicted Prophages Induced in C. beijerinckii

*C. beijerinckii* was the most widely induced bacterium tested (Figure 2B) and led to the largest increase in VLPs (Figure 2B). Several distinct putative prophages were predicted on the genome of our strain of *C. beijerinckii* by VirSorter and PHASTER (Appendix A), more than any of our other bacterial strains (Appendix A). Due to the abundance of VLPs produced by *C. beijerinckii* induction, we were able to obtain enough viral DNA (vDNA) to perform both PCR and shotgun sequencing. This allowed us to investigate which prophages found in the bacterial genome were being induced in *C. beijerinckii* for each treatment of interest.

We increased in silico prophage prediction on *C. beijerinckii* with VIBRANT [37] and PhiSpy [38] to ensure no potential prophages were missed for primer design (Figure 3A). Three prophage regions were scored as complete based on our scoring system: ‘complete’ genome status was determined with three or more tools predicting the region, a lysogeny module, and at least three other modules; ‘uncertain’ genome status was determined when at least two prophage prediction tools identified the region, having less than four modules, and one of the following ‘head’, ‘tail’ or ‘lysis’ morphogenesis modules; and ‘incomplete’ if predicted by just one tool (Figure 3B). To determine which prophages were being induced, we designed PCR primers for all the complete (P1, P2, and P3) and uncertain regions (P4, P5, P7, and P10), as well as a bacteria-specific primer for the *C. beijerinckii dnaA* gene (Appendix A). We reran prophage inductions for ciprofloxacin, mitomycin, norfloxacin, and ampicillin at 2, 1, 10, 0.1 μg mL^−1^, respectively. Primers specific for prophage region P3 amplified DNA in all our treatments, and primers specific for prophage region P1 only amplified in the mitomycin and ampicillin treatments (Figure 4A). None of the other predicted regions were amplified (Figure 4A). P3 and P1 regions were amplified in controls, due to background spontaneous induction that occurs over long growth-curves. We confirmed it is not bacterial contamination, as all vDNA was negative for the bacterial *dnaA* gene (Figure 4A).

In addition, we performed shotgun metagenomics on the extracted vDNA used in each PCR reaction. These qualitative data confirm the PCR detected prophages, and that no prophages were missed during primer design or by prophage detection tools. Normalized read coverage increased within induced prophages regions P1 and P3 (>50 fold), relative to the rest of the bacterial genome in all treatments (Figure 4B, Appendix A). The negative PCR reaction of P1 for the ciprofloxacin and norfloxacin treatments may be due to the limit of detection of our PCR. This is supported by the fact that read coverage of P3 was always higher than in P1 (Appendix A), indicating its induction is likely less productive. Our shotgun metagenomics required an amplification step before sequencing and is therefore not quantitative, but for all treatments except ampicillin, read coverage increased in treatment compared to control (Appendix A), supporting true prophage induction.

We confirmed our approach using prophage induction of the previously reported inducible prophage found in *B. longum* with 2 mM hydrogen peroxide (Appendix A) [58]. Whole genome sequencing of vDNA from *B. longum* indicate that our predicted prophages P4 and P6 (Appendix A) are being induced (Appendix A). The P6 prophage corresponds to the previously reported inducible prophage Binf4 [58]. Our P4 prophage corresponds to two prophages predicted by Ventura et al. [58] (Binf2 and Binf3). We detected induction of prophage P4, which was not detected by Ventura et al. [58] as their primers were designed for complete circularized phage DNA [58] but Binf2 and Binf3 seem to correspond to one large prophage rather than two smaller complete phages.

## 4. Discussion

The gut is an environment in which microorganisms are constantly exposed to medications, whose consumption is on the rise [2,3]. Here, we set out to better understand the role of medications on the gut bacteriophage community, an often-overlooked member of the gut microbiota. Twelve medications from multiple classes were screened to explore their role in prophage induction on eight bacterial lysogens from the human gut. We show that bacterial growth inhibition by these medications leads to prophage induction in at least 55% of cases.

Community-level studies of medications in the gut have shown they are correlated with alterations in bacterial diversity [10,11,12]. One possible explanation for these differences in bacterial diversity can be explained by the direct antibacterial activity of these compounds. For example, NSAIDs, such as diclofenac, have been shown to have an inhibitory effect on bacteria through DNA replication interference [47] similar to quinolones [48]. This is further illustrated in a recent study identifying that NSAIDs had the largest impact on the gut microbiota in a large cohort of healthy adults exposed to a variety of xenobiotics [10]. Chemotherapy medication, fludarabine [50] showed similar inhibition. Ibuprofen for its part was shown to inhibit *Staphylococcus aureus* in a larger screen of six unrelated bacteria [49]. More recently, Maier et al. [13] expanded the study of gut isolates to a large-scale screen of 1000 medications against 40 human gut isolates to understand the direct connection between medications and antibacterial activity. They concluded that 24% of non-antibiotic medications were capable of inhibiting growth of at least one bacterium at concentrations commonly found in the gut. We found a much higher rate of bacterial growth inhibition by non-antibiotics medications, supporting their predictions that increased concentrations would lead to increased antibacterial activity [13] as we often tested concentrations 10-fold higher. Using the same *B. caccae* isolate (*B. caccae* ATCC 43185), we found diclofenac, ibuprofen, tolmetin, busulfan, and acetaminophen to inhibit growth only at concentrations higher than tested by Maier et al. [13]. Yet, the concentrations we tested remain biologically relevant according to our estimations of colonic concentrations.

We also conclude that bacterial growth inhibition resulting from these medications is species-specific. In contrast with Maier and colleagues, who found 11 drugs that led to growth inhibition in all bacteria tested [13], we did not identify “universal” growth inhibitors. Mitomycin, which is often used to detect inducible prophages, was the most effective medication, inhibiting growth in seven isolates. It is important to note that the concentration for mitomycin induction ranged from 0.01–100 µg mL^−1^, and two of our bacteria with inducible prophages were not induced by mitomycin. This could explain the reported underestimation of lysogeny in communities or isolates [59]. Ciprofloxacin, a common replacement for mitomycin in prophage induction experiments, unexpectedly inhibited only three bacterial isolates when given at 2 µg mL^−1^, including *E. coli*, which is known to be inhibited by ciprofloxacin at lower concentrations [60]. This low effect of ciprofloxacin could be explained by the low concentrations tested, as seen in previous studies [13,60,61]. All bacteria were inhibited at higher concentrations of ciprofloxacin, but we report prophage induction for only one (*C. scindens*) (Appendix A).

Collectively, our data support the role of drugs inhibiting bacterial growth in a species-specific manner, which can alter the bacterial diversity of the human gut. We further explored if this growth inhibition could lead to prophage induction, thereby compounding unintended consequences of exposure to these drugs on the gut microbiota.

The antimicrobial activity found in our study was strongly linked to prophage induction of lysogens. VLP production increased in 84% of cases where there was bacterial growth inhibition, and 50% of those increases were statistically significant. Importantly, these increases are not resulting only from antibiotics, previously reported to be prophage inducers, but also from non-antibiotic medication, which have not been reported as prophage inducers. Medications tested included common over-the-counter drugs like acetaminophen and ibuprofen, whose effects on the gut virome have not been reported. Ten of the twelve drugs tested led to prophage induction, and the two drugs for which there was no induction, we nevertheless report an increase in VLPs, suggesting that these compounds can still impact the gut virome.

A limitation to our study was the preliminary screening for bacterial growth inhibition before counting VLPs. First, it is likely that some of our isolates contain prophages inducible by conditions or compounds we have not tested here. For example, we were not able to induce *B. longum*, a strain reported to contain a prophage inducible by hydrogen peroxide [58], with any of our compounds. We thus tested our *B. longum* strain with hydrogen peroxide and saw a significant increase in VLPs without bacterial growth inhibition (2% decrease; Appendix A). In addition, we further quantified VLPs in treatments that were close to our cut-off for bacterial growth inhibition: *C. beijerinckii* (AUC13) exposed to busulfan and *C. scindens* exposed to ibuprofen (AUC7) led to significant increases in VLPs (Appendix A). Thus, by using bacterial growth inhibition as a preliminary screen, our approach leads to a conservative detection of inducible prophages and we are likely underestimating prophage induction by our drugs.

Lastly, epifluorescence microscopy quantification of VLPs does not allow the direct observation of phages. It is thus possible that our VLPs may not be true phages and correspond to other tightly packaged DNA, or membrane vesicles and gene-transfer-agents [62]. Due to superinfection immunity provided by the prophage to the host, we cannot proceed with plaque assays to confirm they are infectious phages. To partly address this concern, we extracted and sequenced the vDNA from our *C. beijerinckii* induction experiments and were able to confirm that the VLPs were indeed true phages induced from within the lysogenic bacterial chromosome.

In our study, we show that a wide range of medication can alter the interactions between phages and bacteria in the gut through prophage induction. The species-specific response to these compounds and resulting differential prophage induction patterns suggest distinct mechanisms of induction, which remain to be investigated. Importantly, such prophage-mediated responses to medications could explain the correlations observed between medication and alterations in the gut phage community [12,14]. Going forward, it will be necessary to tease apart the direct effects of these medications on prophage induction in the gut. Co-culturing bacterial isolates in vitro or using gnotobiotic mouse models, as well as simulated gut communities, will be essential to evaluate the role these species-specific responses have on the gut microbial community, and will allow comparisons with other community-level perturbations such as an inflamed gut environment as in Crohn’s disease [27]. Investigating the downstream consequences of the increased phage abundance and resulting pressure on gut bacterial communities will also help understand the role of prophages in the gut microbiome and their importance for human health.

## Figures and Tables

**Figure 1 viruses-13-00455-f001:**
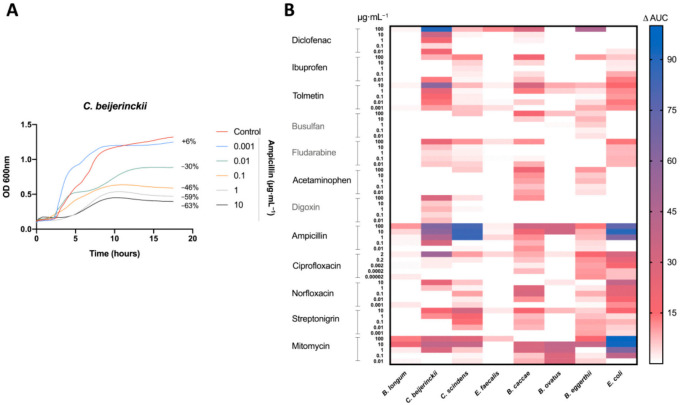
Antimicrobial activity of drugs on human gut isolates: (**A**) Representative growth curve of *C. beijerinckii* (mean OD_600 nm_ measurements of *n* = 3) with ampicillin treatment and DMSO (control). Percent difference in AUC (treatment to control) labelled for each treatment. (**B**) Heatmap of the percentage change in the AUC of all five treatments for each drug compared to the control (DMSO) for all tested bacteria. All drugs were dissolved in DMSO. Control consisted of DMSO at a 1% final concentration. Treatments repeated with a *n* = 3.

**Figure 2 viruses-13-00455-f002:**
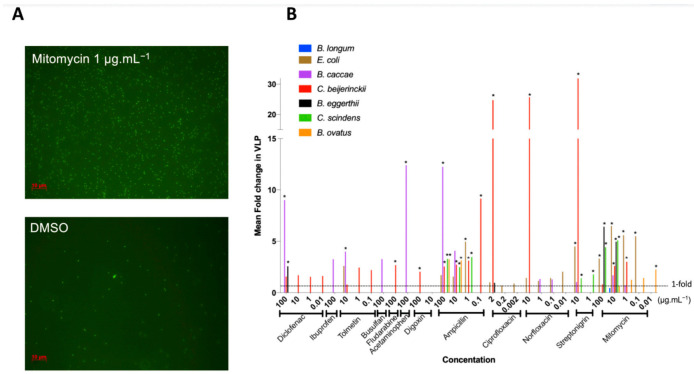
Fold increase in virus-like-particles (VLPs) from antimicrobial activity of drugs: VLPs were counted in drug treatments that resulted in an AUC15. Fold increase in VLPs was obtained by comparing treatment VLP abundance relative to control VLP abundance. (**A**) Representative images of epifluorescence microscopy of SYBR Gold-stained VLPs at 1000X magnification of *C. beijerinckii*. (**B**) Fold increase in VLPs resulting from bacterial growth inhibition by all the drugs tested: mean increase in VLPs (*n* = 3) per treatment compared to DMSO control. * represents *p* < 0.05, Dunnett’s multiple comparison test between treatment and control (*n* = 3).

**Figure 3 viruses-13-00455-f003:**
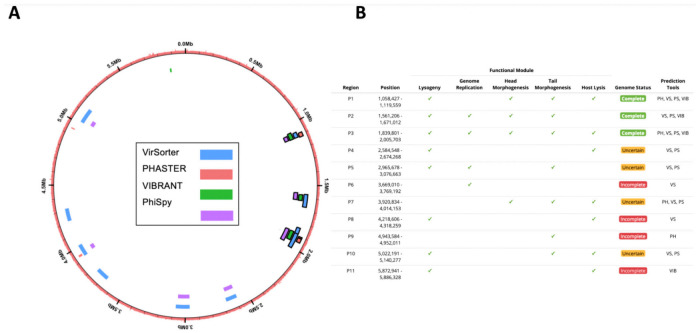
In silico computational prophage prediction of *C. beijerinckii*: Prophages were predicted using PHASTER web-server (PH), VirSorter (VS), PhiSpy (PS), and VIBRANT (VIB) predictive software. (**A**) Predicted prophage regions located within the bacterial genome, color-coded according to the software predictive tool used. Complete prophages have black outline. (**B**) Regions with overlap were merged into 11 predicted prophages P1–11. ORFs were aligned to the prokaryotic virus orthologous groups (pVOG) database using HMMER. The following functional modules were used to classify prophage region completeness: lysogeny (integrases, repressors), genome replication (helicases, ssDNA binding proteins, endonucleases), head morphogenesis (terminases, portal proteins, capsid proteins), tail morphogenesis (tail fiber genes, tail tape measure genes), host lysis (holins, lysins). Three prophage regions were scored as complete based on our scoring system: ‘complete’ genome status was determined with three or more tools predicting the region, a lysogeny module, and at least three other modules; ‘uncertain’ genome status was determined when at least two prophage prediction tools identified the region, having less than four modules, and one of the following ‘head’, ‘tail’ or ‘lysis’ morphogenesis modules; and ‘incomplete’ if predicted by just one tool.

**Figure 4 viruses-13-00455-f004:**
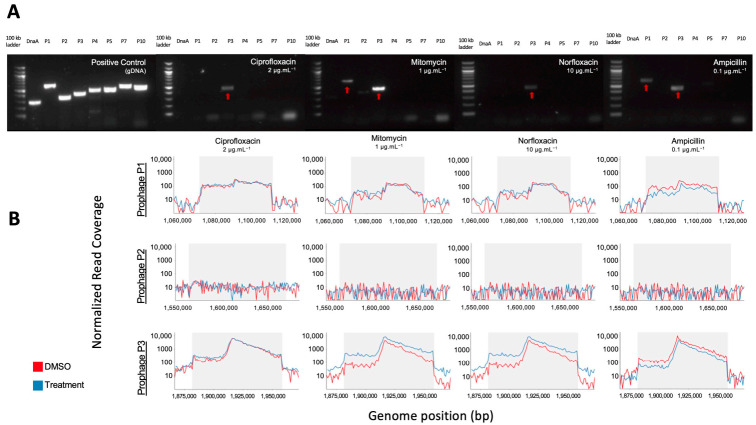
PCR and shotgun sequencing of extracted VLPs from *C. beijerinckii*. (**A**) Agarose gel electrophoresis of PCR products from bacterial DNA and vDNA after exposure to ciprofloxacin (2 µg mL^−1^), mitomycin (1 µg mL^−1^), norfloxacin (10 µg mL^−1^), and ampicillin (0.1 µg mL^−1^) (left to right). Each lane corresponds to one predicted prophage region (P1,P2,P3,P4,P5,P7,P10), the conserved bacterial *dnaA* gene, or a 100 kb ladder. Amplification of P1 and P3 regions show their prophage induction with the corresponding treatment. (**B**) Representative mapping of shotgun sequenced vDNA reads to the genome of *C. beijerinckii* with read coverage increasing within the genome position of predicted complete prophages P1 (**top**), P2 (**middle**) and P3 (**bottom**) for each treatment shown above in the gel electrophoresis. Coverage increased >50× relative to the bacterial genome for prophage regions P1 and P3, but not for P2.

**Table 1 viruses-13-00455-t001:** Collection of lysogenic bacterial isolates tested for inducible prophages.

Phylum	Bacteria	Gram	Accession/Assembly	Isolated	Media
Actinobacteria	*Bifidobacterium longum* subsp. *infantis ATCC 15697*	+	NC_011593	Infant Intestine	BHI w/hemin
Firmicutes	*Clostridium beijerinckii* ATCC 51743	+	GCA_000016965.1	Likely Soil	ABB
	*Clostridium scindens* 32-6-S 4 CNA AN	+	N/A	Human Feces	ABB w/hemin
	*Enterococcus faecalis* TUSoD Ef11	+	NZ_ACOX02000011	Human Oral	BHI
Bacteroidetes	*Bacteroides caccae* ATCC 43185	−	AAVM00000000	Human Feces	TSB
	*Bacteroides ovatus* 3_8_47	−	ACWH00000000	Human Colon biopsy	TSB
	*Bacteroides eggerthii* 1_2_48	−	ACWG00000000	Human Colon biopsy	BHI w/hemin
Proteobacteria	*Escherichia coli* K12 ATCC 25404	−	N/A	Human Feces	BHI

**Table 2 viruses-13-00455-t002:** Medication concentrations and estimated colon concentrations. Estimated colon concentrations were calculated based on oral dose, bioavailability, and volume of average colon (Appendix A). Mitomycin estimated colon concentration was not calculated as it is taken intravenously. NSAID: Non-steroid anti-inflammatory drug.

Type of Agent	Drug	Mechanism of Action	Estimated Colon Concentration (µg/mL)	Tested Concentrations (µg/mL)
Antibiotic	Ampicillin	β-lactam: Cell wall synthesis inhibition	44.56–3565.06	100, 10, 1, 0.1, 0.01
	Ciprofloxacin	Fluoroquinolone: Bacterial DNA gyrase and topoisomerase	106.95–1247.77	2, 0.2, 0.02, 0.002, 0.0002
	Norfloxacin	Fluoroquinolone: Bacterial DNA gyrase and topoisomerase	427.81–998.22	10, 1, 0.1, 0.01, 0.001
	Streptonigrin	Aminoquinone: Bacterial DNA and topoisomerase	0.10–0.19	10, 1, 0.1, 0.01, 0.001
	Mitomycin	DNA Cross Linker	-	100, 10, 1, 0.1, 0.01
NSAID	Diclofenac	Analgesic, antipyretic, and anti-inflammatory	44.56–66.84	100, 10, 1, 0.1, 0.01
	Ibuprofen	Inhibitor of COX	106.95–427.81	100, 10, 1, 0.1, 0.01
	Tolmetin	tNSAID heteroaryl acetic acid derivative	35.65–1048.13	10, 1, 0.1, 0.01, 0.001
Chemotherapy	Busulfan	Alkylating agent-Alkyl sulfonate	1069.52	100, 10, 1, 0.1, 0.01
	Fludarabine	Inhibits DNA Synthesis	7–7.49	100, 10, 1, 0.1, 0.01
Mild Analgesic	Acetaminophen	Not well known	0.00–312.83	100, 10, 1, 0.1, 0.01
Cardiac	Digoxin	Na^+^/K^+^ pumps	0.07–0.13	100, 10, 1, 0.1, 0.01

## Data Availability

Sequence datasets at BioProject PRJNA705204.

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
