# Peer review of "Common Oral Medications Lead to Prophage Induction in Bacterial Isolates from the Human Gut"

_viruses, 2021, doi:10.3390/v13030455_

Round 1
Reviewer 1 Report
The study presented herein examines the ability of many common medications to induce prophage from numerous gut commensals. This work is foundational, as while the effects of xenobiotics on prophage induction has been extensively studied in pathogens, commensal gut bacteria have largely been ignored. Unsurprisingly, they found that specific strains were more susceptible and they then characterized the genomes of the predicted prophages.
The dynamics of phage communities and prophage induction in commensal gut bacteria is so understudied that any study that systematically characterizes these dynamics is of great importance. While I recognize that they provide no mechanism or explanation to why B. caccae and C. beijerinckii induce more prophages than the other strains, I also don't care, because they set up the platform for other researchers to examine their data and study the precise mechanisms in each of these bacteria (or in vivo in the gut virome) later.
Therefore, I think this study is of particular significance to the community now and recommend it for publication.
Minor notes:
Supplementary Figure 2A is more informative than Figure 2B and should be switched.
Figure 1B and Figure 2B: These heatmaps are not as helpful as they could be, and should be colored using a continuous scale rather than a binned scale.
Reviewer 2 Report
The manuscript by S. Sutcliffe et al presented that common oral medications could lead to prophage induction. These findings are quite interesting and show that host-targeted medications may alter the human gut microbiome through increased prophage induction. However, the inductions were tested on individual bacterial isolate. While in human gut, many different types of bacteria exist in population. It would be better to have additional tests to check if there will be any difference in prophage induction between an isolate and the co-culture mix of the isolate with other types of bacteria. Other points:
- Figure 3B: The blue and red color curves are almost identical. How to see the difference between the DMSO control and the treatment?
- Table 1: One strain ATCC51743 is “likely" isolated from soil. Not from human gut? If not, why chose this strain?
Reviewer 3 Report
Review of Manuscript “Common Oral Medications Lead to Prophage Induction in Bacterial Isolates from the Human Gut” by Sutcliffe et al..
With consumption of oral medications constantly rising, the effect of these medications on the gut bacterial environment is of major interest. In their present study the authors have addressed a possible induction of prophages hosted by different bacteria through application of five different medication classes in vitro. These medications included non-steroidal anti-inflammatory agents, chemotherapeutics, analgesics, cardiac agents and antibiotics. They could demonstrate prophage induction for all medication classes under investigation, with the host-targeted medications leading to a rather species-specific induction as compared to antibiotics.
In general the study is well performed, the results are clearly presented and the conclusions drawn by the authors are warranted by their results. However, some major points listed in detail below, which mainly concern the concentrations of some of the antibiotics used in the experiments, should be addressed in a revised version of the manuscript.
Major points:
1) The low maximum concentrations used for the antibiotics Ciprofloxacin and Norfloxacin represent major limitations of the study, since these are several logs below the estimated colon concentrations after oral application (table 2). As stated in the Materials and Methods section, the utilization of these rather low concentrations was due to the limited solubility of the antibiotics in DMSO. Why did the authors not consider using other solvents such as for example water at slightly acidic conditions for Ciprofloxacin? Using higher concentrations in the range of the estimated colon concentrations might have considerably improved the percentages of prophage induction for these antibiotics.
2) Another shortcoming already raised by the authors in the discussion section (lines 409 to 411) is the determination of VLPs only for those compounds and compositions, where at least a 15% bacterial growth inhibition was observed. By this sequential screening the study might have missed a certain percentage of prophage induction events. It would be interesting to obtain the fold increase in VLPs for the complete set of medications conditions and bacterial strains.
Minor points:
1) Results section 3.1, lines 216 to 221: redundant with Materials and Methods section.
2) Fig. 2A: please specify the sample used as a representative image (bacterial strain).
3) Criteria for scoring prophage regions as “complete“, “uncertain“ or “incomplete“, lines 293 to 296 and legend fig. 3B (lines 317 to 321): The complete set of criteria can only obtained when merging the information obtained from the two sections. Please provide all the criteria in either section.
4) Fig. 3B: not readable in printout and poor resolution on screen.
5) Abbreviation WGS (probably whole genome sequencing) should be introduced.
Round 2
Reviewer 2 Report
The manuscript has been revised. Some of my previous concerns are explained. It is now suitable for publication.